# Inflamma-miRs Profile in Myelodysplastic Syndrome Patients

**DOI:** 10.3390/ijms25126784

**Published:** 2024-06-20

**Authors:** Paola Montes, Iryna Rusanova, Elena Cornejo, Paloma García, Ana Guerra-Librero, Mª del Señor López, Tomás de Haro, Germaine Escames, Darío Acuña-Castroviejo

**Affiliations:** 1Centro de Investigación Biomédica, Departamento de Fisiología, Facultad de Medicina, Instituto de Biotecnología, Parque Tecnológico de Ciencias de la Salud, Universidad de Granada, 18016 Granada, Spain; paola.montes.sspa@juntadeandalucia.es (P.M.); aguerit@ugr.es (A.G.-L.); gescames@ugr.es (G.E.); 2UGC de Laboratorios Clínicos, Hospital Universitario Clínico San Cecilio, 18016 Granada, Spain; msenor.lopez.sspa@juntadeandalucia.es (M.d.S.L.); tomas.haro.sspa@juntadeandalucia.es (T.d.H.); 3Departamento de Bioquímica y Biología Molecular I, Facultad de Ciencias, Instituto de Biotecnología, Parque Tecnológico de Ciencias de la Salud, Universidad de Granada, 18016 Granada, Spain; irusanova@ugr.es; 4Centro de Investigación Biomédica en Red Fragilidad y Envejecimiento Saludable (CIBERFES), Instituto Biosanitario de Granada (Ibs.Granada), Hospital Universitario San Cecilio, 18016 Granada, Spain; 5Instituto Biosanitario de Granada (Ibs.Granada), Hospital Universitario Clínico San Cecilio, 18016 Granada, Spain; 6UGC de Hematología y Hemoterapia, Hospital Universitario Clínico San Cecilio, 18016 Granada, Spain; mariae.cornejo.sspa@juntadeandalucia.es (E.C.); paloma.garcia.martin.sspa@juntadeandalucia.es (P.G.)

**Keywords:** myelodysplastic syndrome (MDS), microRNAs (miRNs), oxidative stress, inflammatory cytokines, 5-azacitidine (5-AZA)

## Abstract

Etiological factors involved in myelodysplastic syndrome (MDS) include immunologic, oxidative stress and inflammatory factors, among others, and these are targets for microRNAs (miRNs). Here, we evaluated whether some miRNs may affect tumor development comparing untreated and 5-azacitidine (5-AZA) MDS-treated patients. Peripheral blood samples were collected from 20 controls and 24 MDS patients, and selected miRNs related to redox balance and inflammation (inflamma-miRs), including miR-18a, miR-21, miR-34a and miR-146a, were isolated and measured by quantitative real-time polymerase chain reaction (qRTPCR). A differential expression profile of miRNs was detected in untreated MDS patients and the 5-AZA group. Inflammation increases miRNs and, specifically, miR-18a, miR-21 and miR-34a were significantly overexpressed in untreated MDS, compared to controls. However, we did not observe any miRN profile alteration during the progression of the disease. On the other hand, 5-AZA treatment tends to restore miRN expression levels. Relating to prognostic risk factors, high-risk MDS groups (high Revised International Prognostic Scoring System (IPSS-R), high cytogenetic risk, high molecular risk (HMR) mutations) tended to be related with higher expression levels of miR-18a and miR-34a. Higher miRN expression is correlated with lower glutathione peroxidase activity, while they are related with a higher profile of pro-inflammatory cytokines (IL-2, IL-6, IL-8, TNF-α). Although our study was limited by the low number of MDS patients included, we identified miRN deregulation involved in MDS development that could regulate redox sensors and inflammatory responses. Finally, 5-AZA treatment is related with lower miRN expression levels in MDS patients.

## 1. Introduction

Myelodysplastic Syndromes (MDSs) are hematological disorders with an elevated rate of mortality in the aged population, in which 1 out of 3 patients progress towards acute myeloid leukemia (AML). The course of the disease is highly variable, and, therefore, a recent classification and several prognostic score systems have been incorporated for diagnosis in daily clinical practice [1,2]. In this line, therapies differ between risk groups, in which higher-risk patients are treated with hypomethylating agents such as 5-azacitidine (5-AZA) [3].

Factors determining the pathogenesis and progression of MDS have not been fully elucidated. Among them, oxidative stress plays an important role in the tumorigenesis, as it has been observed in a wide variety of solid and hematological tumors [4]. Recently, our group demonstrated the implication of this biological mechanism in the MDS development and progression of the disease [5]. Oxidative stress has been involved also in the aberrant expression of microRNAs (miRNs), which regulate cellular events such as proliferation, apoptosis, metastasis, and adaptation to hypoxia [6,7]. In turn, miRNs regulate the expression of key components of the cellular antioxidant response. According to this, numerous studies have revealed a set of miRNs that are ROS-sensitive, regulating their transcription. Moreover, the dysregulation of both redox signaling and miRNs has been associated with tumor-promoting inflammation [8].

The precise functions and molecular mechanisms of miRNs/redox connection are still not clear, and the possible interplay between redox regulation and cancer processes is currently being studied [9]. Several antioxidant systems, including the superoxide dismutase (SOD)/catalase (CAT) system, are targets of miRNs (miR-21 and miR-146a, respectively), interfering with their antioxidant functions [10,11]. Related to MDS, several studies have provided evidence of an altered miRN expression profiling in cancer pathogenesis with a potential clinical utility in the diagnosis, genetics changes and defining risk groups [12].

An altered cellular redox status and cytokine expression have been observed in MDS patients [5]. To date, however, no information explains the interplay between oxidative stress parameters, inflammation and miRNs in the pathogenesis of MDS patients. So, we considered it worthwhile to examine for the first time the potential interplay between miRNs and MDS, and the possible impact of the hypomethylating agent 5-AZA treatment on these responses in MDS patients.

## 2. Results

### 2.1. Differential Patterns of miRN Expression in MDS Patients

Relative expression of miR-18a-5p, hsa-miR-21-5p, hsa-miR-34a-5p and hsa-miR-146a-5p was analyzed using hsa-miR-30b-5p as an endogenous control because its expression did not vary in MDS patients and controls [13]. MDS untreated patients had significantly higher levels of miR-18a (*p* = 0.012), miR-21 (*p* < 0.001) and miR-34a (*p* = 0.019) compared to the controls (Figure 1A–C). This increment was significantly higher in the 5-AZA treated group for miR-21 and miR-34a (Figure 1B,C). Although miR-146a expression tends to increase in the MDS untreated patients, no significant differences were observed in the groups (Figure 1D). Moreover, compared to the untreated patients, the 5-AZA group tends to decrease the expression of the four studied miRNs. The treatment with 5-AZA, however, did not recover the expression to the levels of the control group (Figure 1).

### 2.2. miRN Expression and Risk Factors in MDS Patients

To explore if the changes observed in the miRN profile from untreated MDS patients are conserved during the disease progression, we categorized MDS patients according to the recent WHO classification, as early stage (ES, *n* = 13; <5% bone marrow blasts) and advanced stage (AS, *n* = 6; >5% bone marrow blasts). Differences between both stages of the disease in the miRN expression profile, were not detected (Figure 2).

We next analyzed every relationship between miRN expression and the clinical characteristics, such as IPSS-R score, cytogenetic risk and mutational profile. For this evaluation, untreated MDS patients were divided into two groups according to different categories: IPSS-R: high risk (*n* = 9) vs. low risk (*n* = 10); cytogenetic risk: favorable (good karyotypes, *n* = 15) vs. unfavorable (intermediate, poor and very poor karyotypes, *n* = 4); molecular risk: high risk (presence of at least one mutation in any of high molecular risk (HMR) genes (*TP53*, *ETV6*, *ASXL1*, *RUNX1*, *EZH2*) [14], *n* = 12) vs. low risk (absence of mutations in HMR genes, *n* = 6); and mutational score (number of total mutations present in the dysplastic clone (≤2 total mutations, *n* = 14) vs. (>3 total mutations, *n* = 4).

Based on the IPSS-R prognostic stratification, we observed that the expressions of miR-18a and miR-34a tend to increase in MDS patients with a high IPSS-R risk compared with those with a low risk of progression (Figure 3A,C). miR-21 and miR-146a expression levels, however, did not differ between both groups (Figure 3B,D).

When we analyzed miRN expression based on cytogenetic risk, we highlighted the significant increase in miR-21 (*p* = 0.01) and miR-146a (*p* = 0.02) levels in the MDS group with high cytogenetic risk compared to the favorable cytogenetic risk group (Figure 3F,H). miR-18a and miR-34a levels tend to increase in the high-risk group although this increase is not significant (Figure 3E,G).

Finally, we categorized MDS patients concerning molecular risk (high risk, presence of at least one mutation in any of high molecular risk (HMR) genes) vs. low risk (absence of mutations in HMR genes). We observed that miR-21 and miR-34a values tend to be higher in the MDS group that present molecular risk factors than the MDS group with no HMR mutations (Figure 3J,K). 

Also, concerning molecular risk, we evaluated miRN expression between untreated MDS group based on the mutational score (number of total mutations present in the dysplastic clone, ≤2 (*n* = 14) vs. >3 (*n* = 3) total mutations). Similarly, we did not observe differences between patients with a high number of total mutations (>3 total mutations) with respect to those with low mutational score (≤2 total mutations), although miR-21 and miR-34a values tend to be elevated in the MDS group with high risk.

### 2.3. miRN Profile Could Be Related to Redox Signaling and Inflammation in MDS Patients

In our previous work, we analyzed the oxidative stress and inflammatory markers in MDS patients versus the control group [5]. Here, we evaluated possible correlations between the miRN expression profile, the endogenous antioxidant system, and inflammatory markers.

Firstly, we studied the relation between the following antioxidant defense molecules: the glutathione cycle, SOD, CAT, glutathione reductase (GRd), glutathione peroxidase (GPx) and the markers of oxidative damage (lipid peroxidation, LPO and advanced oxidation protein products, AOPP), with the four miRN expressions here analyzed in MDS patients. From all analyses, only a negative correlation between miR-21 and CAT activity (*r* = −0.507, *p* < 0.05) was found (Figure 4A).

Secondly, and taking into an account that in the previous study we analyzed the plasma cytokine levels (IL-1β, IL-2, IL-6, IL-8, IL-10, TNF-α and INF-γ), here we correlated these parameters with miR relative expressions in MDS patients. The negative correlations between miR-18a and TNF-α (r = −0.6471, *p* < 0.005), and between miR-146a and IL-6 levels were detected (*r* = −0.5739, *p* < 0.01) (Figure 4B,C).

## 3. Discussion

miRNs are important regulators of the differentiation and maintenance of hematopoietic stem/progenitor cells, and changes in their expression levels could initiate and promote tumor development. A wide variety of alterations in miRNs from MDS patients has been reported [12,13,15,16]. Thus, we addressed here the possible role of some miRNs in the MDS pathogenesis. 

First of all, our study analyzed risk factors such as *ASXL1*, *SRSF2*, *RUNX1*, *TP53*, which are related to poor prognosis [17]. Based on the molecular characteristics (data reflected in Table 1), the patients were divided according to the presence of high-risk mutations (HRMs). However, the classification based on cellular pathways (RNA splicing, DNA methylation, transcription regulation, etc.) encompasses seven different groups and with the number of patients included in our study, statistical significance is not achieved and it deserves further analysis. The main objective of the present work are the risk factors, because they are directly related to the risk and progression of the disease. Furthermore, we carried out an additional study based on the number of driver mutations [14]. In this context, the relevance of the molecular profiling in the diagnosis, risk assessment and therapeutic decisions, among others, has been emphasized [18]. Our analysis includes both the identification of specific subgroups and the genomic profile using the IPSS-M (Molecular International Prognostic Scoring System) to specify patient risk progression. Based on all this, it is reflected that our data are clinically and biologically useful.

We found that plasma miR-18a, miR-21, miR-34a expression significantly increased in MDS patients compared with controls. This is in accordance with the previous reports that showed an increased expression of miR-18a and miR-21 in the bone marrow and peripheral blood from MDS patients [15]. Moreover, plasma mir-146a expression tends to be higher in the group of MDS patients. It is possible that attention should be paid to this miRN, and the difference could be detected in a greater number of patients. These differences may suggest that tissue expression and plasma levels of this miRN may be differently regulated. Based on our results, upregulation of miR-18a, miR-21 and miR-34a could contribute to the ineffective hematopoiesis in MDS. In this sense, miR-34a has been related to an increase in the apoptosis of bone marrow progenitors [19] and with hematopoietic suppression [20]. No differences were found between genders in the control group or the patients. These results show the possible relationship of miRNs in the pathogenesis of the disease. Although miRN analysis has been carried out on peripheral blood samples and not on MDS cell lineages, the alteration in the miRN profile observed in MDS patients in relation to controls is attributed to the dysplastic cellular process. This observation agrees with the molecular and cytogenetic similarity between analysis of DNA in bone marrow and in plasma in MSD patients published elsewhere [21]. 

Next, we compared the miRN profile expression with the risk of the disease progression according to WHO classification, early (<5% bone marrow blasts) and advanced (>5% bone marrow blasts) MDS stages. We observed that the plasma expression levels of miR-18a, miR-21, miR-34a and miR-146a did not discriminate between both stages of the disease in MDS patients. Other studied miRNs, including miR-422 and miR-617, however, were associated with MDS progression in bone marrow samples [22]. These differences between early MDS subtypes and advanced ones could suggest that altered miRN expression happens from the initial stages of the disease and it may be an event that contributes to the pathogenesis of MDS.

To deepen the role of miRNs on MDS disease, we then investigate the possible variation of their plasma expression levels according to risk prognostic factors of the disease. According to the IPSS-R, we observed that miR-18a and miR-34a expression tended to increase in high-risk (intermediate-/high/very high) compared to low-risk (low and very low) MDS groups, supporting the finding of Choi et al. with miR-21 [23].

Concerning cytogenetic risk, some studies revealed that a karyotype change ((del(5q)/trisomy 8/del(20q)) correlates with a unique profile of miRNs. MDS patients with del(5q) had substantially decreased miRN-146a in the bone marrow cells [16]. We did not study differences between miRN expression levels and an MDS-specific karyotype due to the low number of patients. Here, dividing MDS patients according to cytogenetic risk, we detected a significant increase in miR-21 and miR-146a expression levels in high-risk patients compared with low-risk ones in plasma. 

Analysis of the relationship between the presence of somatic mutations and the levels of circulating miRNs in MDS demonstrated that MDS patient with at least one mutation in HMR genes tend to have higher expression levels of miR-21 and miR-34a than those without HMR mutations. In general, miRN expression increases in MDS patients with the presence of risk prognostic factors, a finding that required further studies that support their use as prognostic biomarkers of risk progression towards AML.

We previously identified an improvement of the intracellular oxidative status in MDS patients, with a significant decrease in the GSSG·GSH^−1^ ratio and glutathione peroxidase (GPx) activity. Moreover, the control of the hydrogen peroxide detoxification was related to the increase in the catalase (CAT) activity, explaining the decrease in the cellular oxidative damage (lipid peroxidation, LPO, and advanced oxidative protein products, AOPP) [5]. According to this, we considered it worthwhile to analyze here miRNs plasma levels with a potential role in oxidative stress. Here, we only found a negative correlation between miR-21 and CAT activity. Because miR-21 inhibits SOD3 preventing hydrogen peroxide production [10], it is suggested that the levels of hydrogen peroxide in MDS patients depend, at least in part, on the elevated miR-21 levels that prevent CAT activity. These effects may favor the oxidative stress condition in MDS patients reported elsewhere [5]. Thus, the dual role of miR-21 on SOD3 and CAT, the primary enzyme systems controlling the source of ROS, probably explains the absence of correlations at secondary levels including GPx and GRd enzymes.

Some shreds of evidences have revealed that miRNs regulate the genes associated with the secretion of different cytokines, including TNF-α, IL-1 and IL-6, among others [24]. We previously observed an upregulation in the IL-2, IL-6, IL-8 and TNF-α levels in plasma samples from MDS patients, reflecting an alteration of the inflammatory signaling pathway [5]. In this sense, miR-146a can suppress inflammatory activity through the downregulation of IRAK1 and TRAF6, resulting in the inhibition of the NF-κB pathway [25,26]. Furthermore, miR-146a downregulates IL-6 production in aging and some pathological conditions [24,25,27]. Together these data may reflect an attempt of miR-146a to prevent further increase in IL-6 in MDS patients. Additionally, we also found a negative correlation between miR-18a and TNF-α levels, whereas miR-18a is highly expressed in glioma and it can affect proliferation, migration and invasion of human glioblastoma cells [28]; an explanation for the connection between these molecules here shown requires additional analysis. 

Finally, our group has also recently described that 5-AZA treatment increases oxidative stress in MDS patients, with a decrease in the catalase activity related to a marked increase in plasma LPO [5]. Analogously, we investigated whether treatment with 5-AZA modified the expression profiles of the miRNs studied. We observed that 5-AZA group decreased all miRN expression compared with untreated MDS patients, although the levels of the control group were not recovered. According to this, low miR-21 expression levels in the serum from MDS patients seem to predict a response to hypomethylating agents [29]. Similarly, treatment with an epigenetic therapy (lenalidomide) decreases miR-34a-3p and miR-34a-5p expression levels in peripheral blood monocytes [30].

## 4. Material and Methods

### 4.1. Study Design

#### 4.1.1. Patients and Controls

A total of 24 myelodysplastic syndrome (MDS) patients (12 males and 12 females; mean age of 70 years) of the San Cecilio’s University Hospital of Granada were included in the study. MDS patients were registered on the Spanish MDS Registry (RESMD), and their subtypes were defined according to the World Health Organization (WHO-2016) [2] as early-stage disease (ES, <5% bone marrow blasts) and advances-stage disease (ASD, >5% bone marrow blasts). The ES subgroup included 2 patients with single lineage dysplasia (MDS-SLD), 7 with multilineage dysplasia (MDS-MLD), 1 with SLD and ring sideroblasts (MDS-RS-SLD), 2 with MLD and ring sideroblasts (MDS-RS-MLD), 2 with ring sideroblasts (MDS-RS) and 2 with isolated del(5q) MDS (del5q). The ASD subgroup involved 4 patients with excess blasts-1 (MDS EB-1), and 4 with excess blasts-2 (MDS EB-2). Based on the revised International Prognostic Scoring System (IPSS-R) all patients were divided into lower-risk disease (IPSS-R very low, low) and higher-risk disease (IPSS-R intermediate, high, very high). The cytogenetic risk categories were divided as follows: Favorable (Very good, −Y, del(11q), Good (normal, del(20q), del(5q) alone or with 1 other anomaly, and del(12p)), Poor (complex with 3 abnormalities, der(3q) or chromosome 7 abnormalities), Very poor (complex with ≥3 abnormalities), and Intermediate (all other single or double abnormalities not listed) (Table 1).

MDS patients were classified according to the established therapy as untreated MDS patients (patients at diagnosis or only with supportive care including erythropoietin, Epo and/or granulocyte macrophage colony-stimulating factor, GM-CSF) or treated MDS patients (patients treated with hypomethylating agents such as 5-AZA from 3 to 7 cycles). Only one MDS patient received chemotherapy treatment and was included within treated patients.

Samples from 20 healthy volunteer donors (control group) were also analyzed. All controls (3 males and 17 females, median age 70 years; range 42–88 years), had normal hemograms and they had no history of neoplastic disease, or previous exposure to chemotherapy drugs, radiation therapy or immunotherapy.

#### 4.1.2. Blood Samples

Peripheral blood samples were collected by peripheral venipuncture from the antecubital vein in MDS patients and controls between 8 and 10 a.m. Blood samples were centrifuged at 3000 rpm for 15 min. Plasma and erythrocytes were separated (erythrocytes were washed twice with cold saline) and were aliquoted and stored at −80 °C for oxidative stress parameter studies. Aliquots of miRNs plasma assays were collected into RNase/DNase-free tubes and immediately aliquoted and frozen at −80 °C until the assays were performed.

#### 4.1.3. RNA Isolation and Quantification of Circulating miRN Levels

The total RNA extraction was performed using a miRNeasy Serum/Plasma advanced isolation kit (Qiagen, Toronto, Canada) according to the manufacturer’s instructions. Briefly, 200 μL of plasma samples was used to extract the total RNA, and an equal volume of denaturing solution was added (20 μL). Four miRNs (miR-18a-5p, hsa-miR-21-5p, hsa-miR-34a-5p and hsa-miR-146a-5p) with a potential role in oxidative stress and inflammation were analyzed. 

#### 4.1.4. Quantitative Real-Time Amplification (qRT-PCR)

Real-Time PCR technique was performed in a Stratagene Mx3005P QPCR System (Agilent Technologies, Barcelona, Spain) according to the manufacturer’s protocol, using TaqMan Fast Advanced Master Mix (2×) and TaqMan probes (20×) (assay names: hsa-miR-18a-5p, assay ID: 478551; hsa-miR-21-5p, assay ID: 477975; hsa-miR-34a-5p, assay ID: 478048; hsa-miR-146a-5p, assay ID: 478399) (Thermo Fisher Scientific, Waltham, MA, USA). The total reaction volume was 20 μL per well. The reaction conditions were enzyme activation, 20 s at 95 °C, denaturation, 3 s at 95 °C (40 cycles), and elongation, 30 s at 60 °C. The levels of plasma miRN expression were normalized to hsa-miR-30b-5p (assay ID: 47995, Thermo Fisher Scientific, Waltham, MA, USA) as a housekeeping gene using the 2^−ΔCt^ and 2^−ΔΔCt^ formulas.

#### 4.1.5. Measurement of the Antioxidative Defense

The evaluation of antioxidant defense molecules (disulfide glutathione, GSSG, and glutathione, GSH) and the activities of the enzymes involved (superoxide dismutase, SOD; catalase, CAT; glutathione reductase, GRd; and glutathione peroxidase, GPx) were measured as described in Montes et al. [5].

#### 4.1.6. Assessment of Cytokine Levels

Determinations of pro-inflammatory cytokines were carried out in the plasma fraction as described in Montes et al. [5].

### 4.2. Statistical Analysis

Relative expressions of miRNs between MDS groups were compared using the nonparametric Mann–Whitney U test or Kruskal–Wallis test. Data are expressed as mean ± SEM. Correlations between miRNs and oxidative stress parameters and cytokines were analyzed using Pearson’s correlation coefficient. GraphPad Prism v. 6.0 (GraphPad Software, Inc., La Jolla, CA, USA) was used to analyze data. *p* < 0.05 was considered statistically significant.

## 5. Conclusions

To summarize, miRNs are known to regulate critical cellular processes and, in the present investigation, miR-18a, miR-21, miR-34a and miR-146a have been related to the MDS pathogenesis, being associated with risk MDS groups, and correlated with pro-inflammatory cytokines. In this sense, miRNs could be used as a possible therapeutic tool, in which the inhibition or downregulation of these miRNs might re-establish the MDS microenvironment. Finally, 5-AZA treatment tends to decrease miRN expression levels in the serum from MDS patients. To further clarify the interpretation of the results, our data reflect changes in miRN expression and a correlation in MDS patients with treatment, but we cannot know at this time whether these changes are due to cause or are an effect of them. In addition, although we correlated here the changes in miRNs with treatment in MDS patients, it should be considered that these miRN fluctuations may also result in cellular changes, which requires further analysis.

### Limitations of the Study

We are aware that the number of patients collected for this study is low, mainly due to the requirements for the inclusion in the study and the relative low percentage of MDS patients, and especially in cases when these patients were divided into subgroups according to risk factors (stage of the disease, IPSS-R, cytogenetic risk and mutational profile). Thus, although we found some tends of the markers measured in our study, it would be possible to detect significant changes by increasing the number of patients.

## Figures and Tables

**Figure 1 ijms-25-06784-f001:**
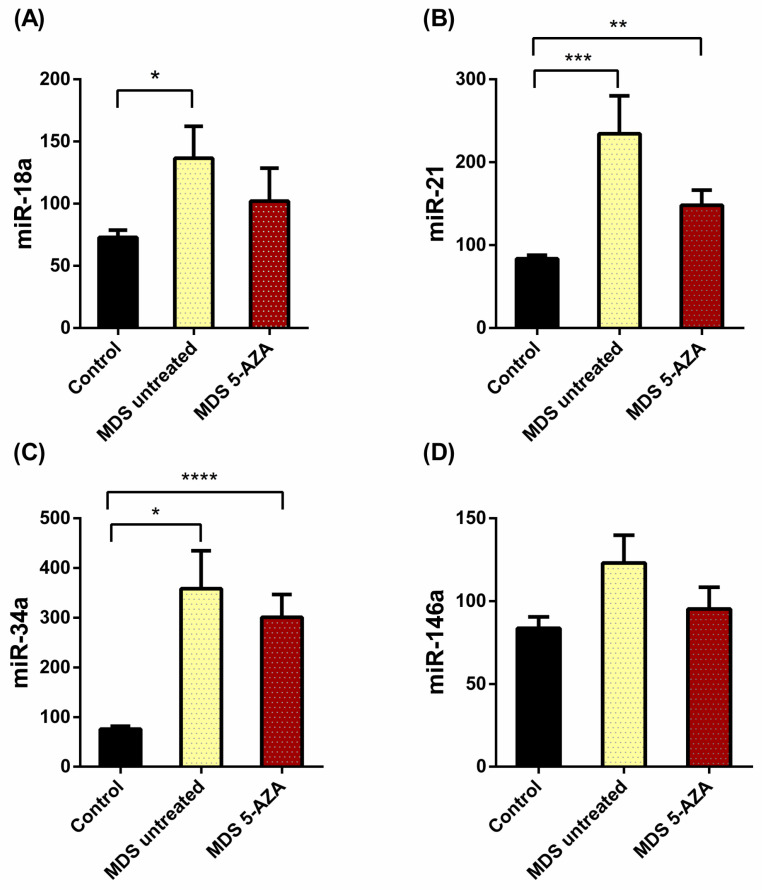
Changes in the relative expression of the miRN transcripts in MDS untreated (*n* = 19) and MDS 5-AZA-treated (*n* = 7) patients. The following miRN levels are represented: (**A**) miR-18a, (**B**) miR-21, (**C**) miR-34a, (**D**) miR-146a. Data are expressed as means ± SEM of miRNs (miR), using miR-30b as an endogenous control in plasma samples of 24 myelodysplastic syndrome (MDS) patients and controls. * *p* < 0.05, ** *p* < 0.01, *** *p* < 0.001, **** *p* < 0.0001 vs. control.

**Figure 2 ijms-25-06784-f002:**
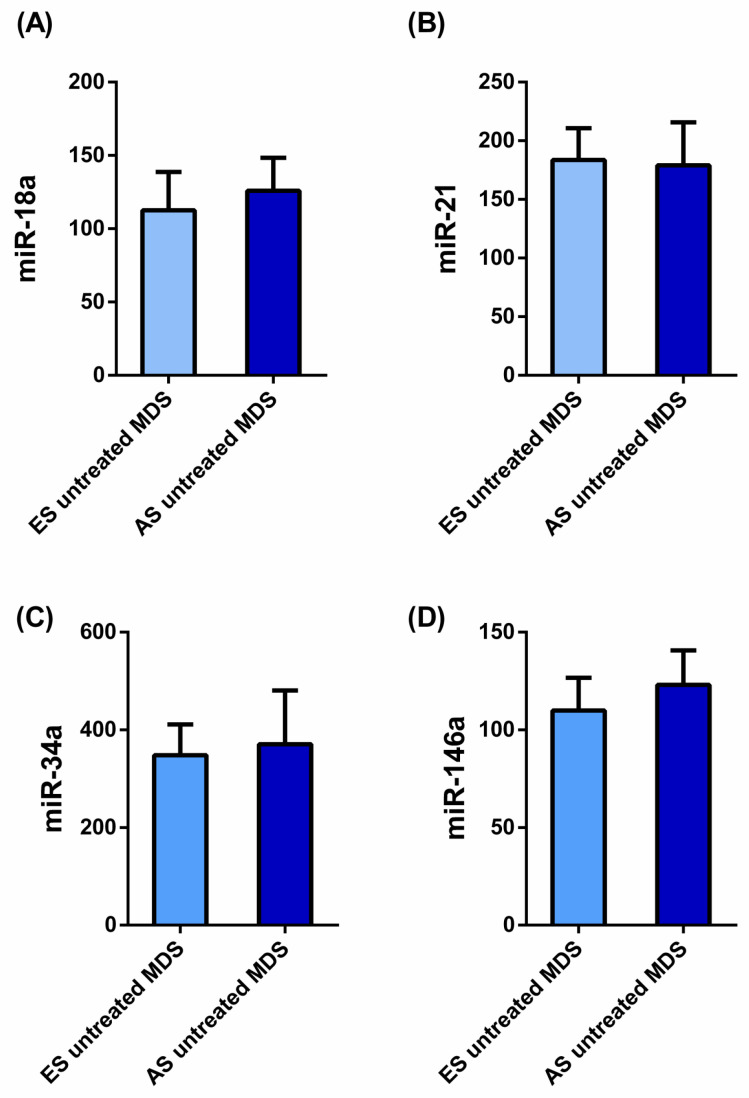
The relative expression of the miRN transcripts in MDS untreated patients (*n* = 19) during the disease progression. The relative expression of (**A**) miR-18a, (**B**) miR-21, (**C**) miR-34a and (**D**) miR-146a is represented. Data are expressed as means ± SEM of miRNs (miR), using miR-30b as an endogenous control in plasma samples of 19 untreated myelodysplastic syndrome (MDS) patients according to the disease progression: Early Stage (ES) untreated MDS (*n* = 13) and Advanced Stage (AS) untreated MDS (*n* = 6). No significant differences were observed (*p* > 0.05).

**Figure 3 ijms-25-06784-f003:**
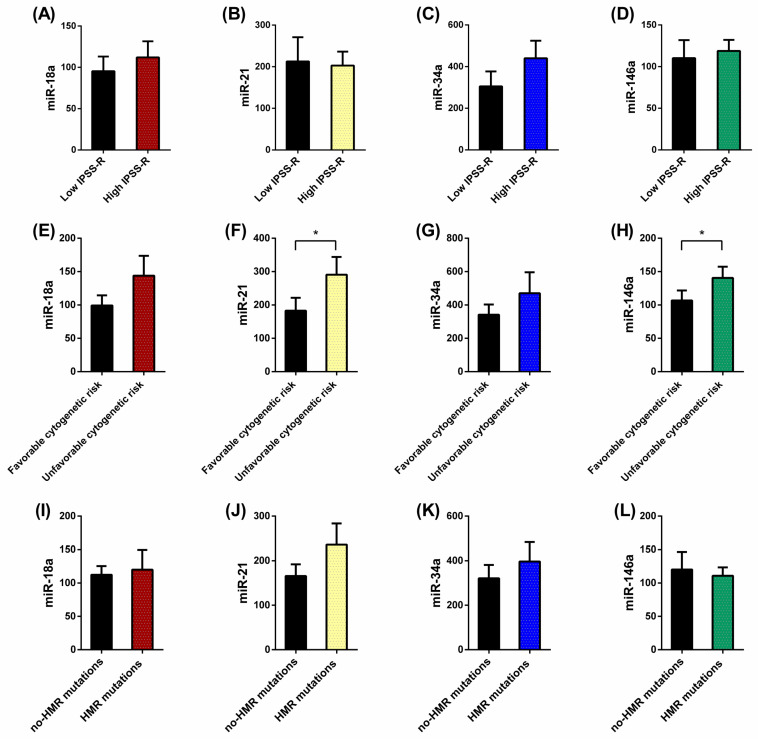
The relative expression of (**A**,**E**,**I**) miR-18a, (**B**,**F**,**J**) miR-21, (**C**,**G**,**K**) miR-34a and (**D**,**H**,**L**) miR-146a, using miR-30b as an endogenous control in plasma samples of 24 patients with myelodysplastic syndrome (MDS), classified according to the following risk prognostic factors: IPSS-R score (**A**–**D**) (high risk, *n* = 9; low risk, *n* = 10), cytogenetic risk (**E**–**H**) (favorable, *n* = 15; unfavorable, *n* = 4); and molecular risk (**I**–**L**) (high risk, *n* = 12; low risk, *n* = 6). The *p* values were calculated by the non-parametric Mann–Whitney test. Data are presented as mean ± SEM. * *p* < 0.05 vs. control.

**Figure 4 ijms-25-06784-f004:**
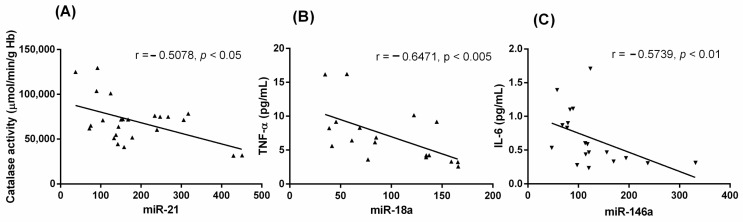
Correlation analysis between (**A**) miR-21 and CAT activity, miR-18a and TNF-α (**B**), and miR-146a and IL-6 (**C**) in MDS patients. Analysis was conducted with the Pearson’s correlation coefficient.

**Table 1 ijms-25-06784-t001:** Clinical characteristics of the 24 MDS patients of the study.

UPN	Age	Gender	WHO-2016	IPSS-R	IPSS-R Risk	Cytogenetic Risk	Mutational Profile	Score Mutational	Molecular Risk	Treatment
1	80	MALE	MDS-SLD (Early Stage)	BR	Low Risk	Good (Favorable)	***ASXL1*** (37%), *U2AF1* (41.8%), *SETBP1* (2.9%)	3	High Risk	Azacitidine
2	42	FEMALE	MDS-SLD (Early Stage)	VL	Low Risk	Good (Favorable)	***ASXL1*** (44.2%), *U2AF1* (40.85%)	2	High Risk	NA
3	51	MALE	MDS-RS-SLD (Early Stage)	BR	Low Risk	Good (Favorable)	*SF3B1* (43.2%), *TET2* (47.2%)	2	Low Risk	NA
4	75	FEMALE	MDS-MLD (Early Stage)	INT	High Risk	Good (Favorable)	***TP 53*** (28.6%), ***ASXL1***(25.4%), *SRSF2* (28.6%), *U2AF1* (30.6%)	4	High Risk	Azacitidine
5	65	MALE	MDS-MLD (Early Stage)	VL	Low Risk	Good (Favorable)	***ASXL1*** (18.3%), ***EZH2*** (37.3%), ***RUNX1*** (9.9%), *TET2* (28%)	4	High Risk	Azacitidine
6	73	FEMALE	MDS-MLD (Early Stage)	INT	High Risk	Poor (Unfavorable)	***TP53*** (76.4%)	1	High Risk	NA
7	71	MALE	MDS-MLD (Early Stage)	VL	Low Risk	Good (Favorable)	*IDH2* (46,6%), *DNMT3A* (45%)	2	Low Risk	NA
							*IDH2* (46.7%), *DNMT3A* (46.5%)	2	Low Risk	Azacitidine
8	78	MALE	MDS-MLD (Early Stage)	VL	Low Risk	Very Good (Favorable)	ND	ND	ND	Supportive care
9	61	FEMALE	MDS-MLD (Early Stage)	VL	Low Risk	Good (Favorable)	Not detected	0	Low Risk	NA
10	73	FEMALE	MDS-MLD (Early Stage)	BR	Low Risk	Good (Favorable)	***ASXL1*** (12.5%)	1	High Risk	Supportive care
11	88	FEMALE	MDS-RS-MLD (Early Stage)	BR	Low Risk	Good (Favorable)	ND	ND	ND	Supportive care
12	80	FEMALE	MDS-RS-MLD (Early Stage)	VL	Low Risk	Good (Favorable)	***RUNX1*** (49.3%), *SF3B1* (28%)	2	High Risk	NA
13	66	FEMALE	MDS *del*(5q) (Early Stage)	INT	High Risk	Good (Favorable)	Not detected	0	Low Risk	Supportive care
14	82	MALE	MDS *del*(5q) (Early Stage)	BR	Low Risk	Good (Favorable)	***ASXL1*** (47.7%), *TET2* (47.3%)	2	High Risk	NA
15	61	FEMALE	MDS SA (Early Stage)	BR	Low Risk	Good (Favorable)	***RUNX1*** (18.8%), *SF3B1* (42%), *JAK2* (5.8%), *TET2* (40.9%)	4	High Risk	Supportive care
16	77	FEMALE	MDS SA (Early Stage)	INT	High Risk	Poor (Unfavorable)	*SF3B1* (40.3%)	1	Low Risk	NA
17	78	MALE	MDS EB-1 (Advanced Stage)	AR	High Risk	Intermediate (Unfavorable)	***ASXL1*** (32.5%), *ZRSR2* (67.4%)	2	High Risk	Supportive care
18	75	MALE	MDS EB-1 (Advanced Stage)	INT	High Risk	Good (Favorable)	***ASXL1*** (11.6%)	1	High Risk	Azacitidine
19	73	FEMALE	MDS EB-1 (Advanced Stage)	INT	High Risk	Good (Favorable)	*WT1* (11.8%)	1	Low Risk	NA
							*WT1* (17.4%)	1	Low Risk	Azacitidine
20	65	FEMALE	MDS EB-1 (Advanced Stage)	MAR	High Risk	Very Poor (Unfavorable)	***TP53*** (2.9%), ***RUNX1***(1.9%)	2	High Risk	NA
21	65	MALE	MDS EB-2 (Advanced Stage)	INT	High Risk	Good (Favorable)	***ASXL1*** (31.1%), *FLT3* (9.3%), *SRSF2* (32.8%), *TET2* (33.3%), *NRAS c.35G>A* (11.8%), *NRAS c34G>A* (5.9%), *ETV6* (2.5%)	7	High Risk	NA
							***ASXL1*** (18%), *FLT3* (1%), *SRSF2* (21.1%), *TET2* (42.7%), *NRAS c.35G>A* (3.9%), *NRAS c34G>A* (1.1%), *ETV6* (3.3%), *CEBPA* (2.7%), *JAK2* (16.4%), *CBL* (10%)	10	High Risk	Chemotherapy
22	66	MALE	MDS EB-2 (Advanced Stage)	MAR	High Risk	Intermediate (Unfavorable)	***ASXL1*** (45.9%), ***RUNX1***(48%), ***EZH2***(94.7%), *NRAS* (11.5%), *FLT3* (2%), *CEBPA* (1.7%)	6	High Risk	Azacitidine
23	83	MALE	MDS EB-2 (Advanced Stage)	MAR	High Risk	Very Poor (Unfavorable)	***TP53*** (70.5%)	1	High Risk	NA
24	72	MALE	MDS EB-2 (Advanced Stage)	AR	High Risk	Good (Favorable)	***ASXL1*** (41.6%), ***RUNX1*** (47.3%), ***EZH2*** (93.3%), *CSF3R* (45.1%), *CSF3R* (5.9%)	5	High Risk	NA

Note: Unique Patient Number (UPN); Myelysplastic Syndrome (MDS); MDS with single lineage dysplasia: MDS-SLD; MDS with multilineage dysplasia: MDS-MLD; MDS with ring sideroblasts: MDS-RS; MDS with SLD and ring sideroblasts (MDS-RS-SLD); MDS with MLD and ring sideroblasts (MDS-RS-MLD); MDS with isolated del(5q): MDS (del5q); MDS EB-1, -2: MDS with Excess of Blasts-1, -2. IPSS-R: Revised International Prognostic Scoring System. Low Risk (Very Low Risk (VLR)/Low Risk (LR); High Risk (Intermediate Risk (INT)/High Risk (HR)/Very High Risk (VHR). Cytogenetic risk: Favorable (Very good (-Y, del(11q), Good (normal, del(20q), del(5q) alone or with 1 other anomaly and del(12p)); Poor (Poor (complex with 3 abnormalities, der(3q) or chromosome 7 abnormalities), Very poor (complex with ≥3 abnormalities)); Intermediate (all other single or double abnormalities not listed). Favorable (Good karyotypes) vs. Unfavorable (Intermediate, Poor and Very Poor karyotypes). Mutational Score: number of total somatic mutations detected in each patient. In Mutational Profile, the Variant Allelic Frequencies (VAFs) are shown in parentheses and genes indicated in bold are High Molecular Risk (HMR) genes. Molecular risk: High Risk (presence of at least one mutation in any of High Molecular Risk (HMR) genes) vs. Low Risk (absence of mutations in HMR genes). NA: Not Applicable; ND: No Data.

## Data Availability

The datasets generated during and/or analyzed during the current study are available from the corresponding author (dacuna@ugr.es) on reasonable request. Materials described in the manuscript will be freely available to any research to use them for noncommercial purposes.

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
