# Peer review of "Inflamma-miRs Profile in Myelodysplastic Syndrome Patients"

_ijms, 2024, doi:10.3390/ijms25126784_

Round 1
Reviewer 1 Report (Previous Reviewer 1)
Comments and Suggestions for Authors
The manuscript has been reasonably revised in terms ot objectives of this study in introduction sectuin and interpretation of obtained results in discussion section.
Comments on the Quality of English LanguageEnglish writing is generraly good.
Author Response
- miRNs parameters were correlated with some proinflammatory and antioxidant defense parameters, which were obtained from previous research. It would be appropriate to mention these parameters in Material and Methods section and cite the reference where it was described how these parameters were determined.
We have included this information in the Material and Methods section (2.1.5. Measurement of the antioxidative defense) and (2.1.6. Assessment of cytokine levels).
The determination of pro-inflammatory cytokines were carried out in the plasma fraction as described in Montes et al. [5].
- Regarding IPSS-R, it is stated that low risk disease is very low, low and intermediate (line 91), and below the Table 1, intermediate risk is high risk disease. It should be uniformed.
revised and corrected
- Line 130, spelling error, RNA isolation instead of RNA solation.
Corrected
- 24 patients were included in the study. It should be stated as a limitation of the research, especially in cases when these patients were divided into subgroups according to risk factors, early-stage disease, advance stage disease……
We have included an statement of this point in the article the section: 6. Limitations of the study
- In Conclusion section, the authors stated… Finally, 5-AZA treatment could restore miRNs expression levels in the serum from MDS patients…. But as it can be seen on the Figure 1, there were only decreasing trend of miRNs parameters in 5-AZA treated group compared to untreated group, but with no significant difference. So, the conclusion should be rephrased.
We have replaced by: “Finally, 5-AZA treatment tends to decrease miRNs expression levels in the serum from MDS patients”
Reviewer 2 Report (New Reviewer)
Comments and Suggestions for Authors
The authors evaluated potential interplay between some miRNs (miR-18a, miR-21, miR-34a and miR-146a) and MDS, and the possible effect of 5-azacitidine (5-AZA) treatment on these responses in MDS patients. They concluded that miR-18a, miR-21, miR-34a and miR-146a have been related to the MDS pathogenesis, being associated with risk MDS groups, and correlated with pro-inflammatory cytokines.
This is interesting study and the results have potential to be clinically and biologically useful.
I have some questions and suggestions to the authors.
miRNs parameters were correlated with some proinflammatory and antioxidant defense parameters, which were obtained from previous research. It would be appropriate to mention these parameters in Material and Methods section and cite the reference where it was described how these parameters were determined.
Regarding IPSS-R, it is stated that low risk disease is very low, low and intermediate (line 91), and below the Table 1, intermediate risk is high risk disease. It should be uniformed.
Line 130, spelling error, RNA isolation instead of RNA solation.
24 patients were included in the study. It should be stated as a limitation of the research, especially in cases when these patients were divided into subgroups according to risk factors, early-stage disease, advance stage disease……
In Conclusion section, the authors stated… Finally, 5-AZA treatment could restore miRNs expression levels in the serum from MDS patients…. But as it can be seen on the Figure 1, there were only decreasing trend of miRNs parameters in 5-AZA treated group compared to untreated group, but with no significant difference. So, the conclusion should be rephrased.
Comments on the Quality of English LanguageMinor editing of English language required.
Author Response
Thanks very much for your comments.
Round 2
Reviewer 2 Report (New Reviewer)
Comments and Suggestions for Authors
The authors responded to all comments.
Comments on the Quality of English LanguageMinor editing of English language required.
Author Response
The manuscript has been revised and English language and typos errors have been corrected
This manuscript is a resubmission of an earlier submission. The following is a list of the peer review reports and author responses from that submission.
Round 1
Reviewer 1 Report
Comments and Suggestions for Authors
The authors examined blood expression levels of miRNAs in patients with MDS and analyzed relationship between miRNA levels and MDS statuses including those after treatment with hypomethylating agents. In this article, however, definite conclusions cannot be drawn because of too many negative data regarding the relationship between miRNA levels and MDS status.
Major comments:
1. The authors examined blood miRNA expression levels and observed higher expression in MDS cohort than healthy control; however, it is unclear whether or not the miRNAs were derived from MDS cell lineages.
2. In this study, significant relationship between the miRNA expression levels and MDS statuses such as after 5 AZA treatment, MDS stage, IPSS-R, and gene mutational score was not observed. Therefore, the miRNA expressions examined in this study cannot be the profile of MDS, much less molecular pathogenesis of MDS.
3. The authors showed negative correlations between miRNA expressions and blood TNF-a and IL-6 levels. However, these cytokines reflect universal inflammation and not characteristic for MDS.
4. The authors also showed negative correlation between miRNA expressions and catalase activity; however, nobody can generalize this result to dysregulation of redox signaling in MDS because of only a single significant result.
Minor comment:
Lines 91-94: Favorable, Very good, etc, should be favorable, very good, for example.
Comments on the Quality of English LanguageEnglish writing is generally good in tjis manuscript.
Author Response
The authors examined blood expression levels of miRNAs in patients with MDS and analyzed relationship between miRNA levels and MDS statuses including those after treatment with hypomethylating agents. In this article, however, definite conclusions cannot be drawn because of too many negative data regarding the relationship between miRNA levels and MDS status.
Major comments
- The authors examined blood miRNA expression levels and observed higher expression in MDS cohort than healthy control; however, it is unclear whether or not the miRNAs were derived from MDS cell lineages.
The miRNA expression was analyzed from peripheral blood samples, not specifically from MDS cell lineages.
- In this study, significant relationship between the miRNA expression levels and MDS statuses such as after 5 AZA treatment, MDS stage, IPSS-R, and gene mutational score was not observed. Therefore, the miRNA expressions examined in this study cannot be the profile of MDS, much less molecular pathogenesis of MDS.
The miRNA expression is obtained from peripheral blood samples; the pool of these miRNAs, although not specifically obtained from MDS cells, also reflect the miRNAs content in these cells.
- The authors showed negative correlations between miRNA expressions and blood TNF-a and IL-6 levels. However, these cytokines reflect universal inflammation and not characteristic for MDS.
From the exclusion criteria, patients with other inflammatory process apart from MDS disease were excluded in the study. So, the blood levels of the proinflammatory TNF-a and IL-6 largely reflect the inflammation in these patients.
- The authors also showed negative correlation between miRNA expressions and catalase activity; however, nobody can generalize this result to dysregulation of redox signaling in MDS because of only a single significant result.
We agree with this comment; however, these data are continuation of a previously published paper in the same group of MDS patients in which we analyzed the redox status of them, confirming the existence of an oxidative stress.
Minor comment:
Lines 91-94: Favorable, Very good, etc, should be favorable, very good, for example.
Corrected.
Reviewer 2 Report
Comments and Suggestions for Authors
This report by Montes et al documented blood plasma levels of 4 miRNAs in 24 MDS patients and 20 normal controls. While some of the findings are of interest, the scope/impact of the study is severely limited by the small sample size as authors so acknowledged.
Major comment
Inclusion of new data from more MDS patients with and without 5-AZA treatment will significantly strengthen the paper. If statistical significance can be achieved, authors could reach a more definitive conclusion affirming the predictive values of plasma miR-18a, miR-21 and miR-34a levels on MDS diagnosis and treatment response.
Minor comments
1. A Table should be organized to document basic conditions of each MDS patient and normal control, with parameters such as: UPN, age, gender, diagnosis, IPSS score, with/without 5-AZA treatment, etc.
2. The number of samples analyzed for each group should be stated clearly in each figure legend.
3. Figure 4 presented correlations between miRNA measurements reported in current study and other measurements reported previously (from the same population of MDS patients presumably). It seems that a large number of correlation analyses were performed (8 parameters associated with antioxidant + 7 cytokines) x 4 miRNAs = 60 correlations?). If that is the case, potential random error could increase significantly. Thus, the alpha value should be adjusted when declaring statistical significance. Some of the three significant correlations presented in Figure 4 might become insignificant if an adjusted alpha values is used. Authors may wish to consult a statistician regarding data analysis and presentation.
Comments on the Quality of English LanguageModerate English editing is recommended.
Author Response
Reviewer 2
This report by Montes et al documented blood plasma levels of 4 miRNAs in 24 MDS patients and 20 normal controls. While some of the findings are of interest, the scope/impact of the study is severely limited by the small sample size as authors so acknowledged.
Major comment
Inclusion of new data from more MDS patients with and without 5-AZA treatment will significantly strengthen the paper. If statistical significance can be achieved, authors could reach a more definitive conclusion affirming the predictive values of plasma miR-18a, miR-21 and miR-34a levels on MDS diagnosis and treatment response.
We agree with this observation. However, the number of MDS patients is too small, and we only could collect the ones entering in the study. The incidence of MDS in Spain is estimated at 4-5 cases per 100,000 persons-year “Garcia-Manero G. Myelodysplastic syndromes: 2015 Update on diagnosis, risk-stratification and management. Am J Hematol. 2015; 90: 831-41. doi:10.1002/ajh.24102”.
Minor comments
- A Table should be organized to document basic conditions of each MDS patient and normal control, with parameters such as: UPN, age, gender, diagnosis, IPSS score, with/without 5-AZA treatment, etc.
According to the reviewer, we have prepared a table that includes the clinical characteristics of the 24 MDS patients of the study (Supplemental Table 1). Characteristics of the control group (age and gender) have been included in the main text, within material and methods section.
- The number of samples analyzed for each group should be stated clearly in each figure legend.
As the reviewer comments, we have indicated in each figure legend (Figure 1, Figure 2, Figure 3 and Figure 4) the number of samples analyzed in each group.
- Figure 4 presented correlations between miRNA measurements reported in current study and other measurements reported previously (from the same population of MDS patients presumably). It seems that a large number of correlation analyses were performed (8 parameters associated with antioxidant + 7 cytokines) x 4 miRNAs = 60 correlations?). If that is the case, potential random error could increase significantly. Thus, the alpha value should be adjusted when declaring statistical significance. Some of the three significant correlations presented in Figure 4 might become insignificant if an adjusted alpha values is used. Authors may wish to consult a statistician regarding data analysis and presentation.
All correlations in the patient group have been made using Graph Pad Prism v. 6.0, as follows: for each miR we correlated with the oxidative stress markers and with the inflammation markers, separately. P value: two tailed, confidence interval: 95%. In all correlation analysis we used alpha value = 0.05.
Round 2
Reviewer 1 Report
Comments and Suggestions for Authors
Although the authors submitted a revised manuscript, the revision is restricting and almost same to former manuscript. Therefor, my evaluation for this study must become same as the previous one.
Major comment
Because of only a few significant results, the miRNA expressions examined in this study cannot be the profile of MDS, and this study does not contribute to the field of MDS molecular investigation.
Comments on the Quality of English LanguageEnglish writing is generally good.
Reviewer 2 Report
Comments and Suggestions for Authors
The Table provided important information of the 24 MDS patients and should be presented as a normal Table, not as a supplementary Table, so that readers will have easy access to the related information.